# Artificial Intelligence-Driven Translation Tools in Intensive Care Units for Enhancing Communication and Research

**DOI:** 10.3390/ijerph22010095

**Published:** 2025-01-12

**Authors:** Sahar Bahrami, Francesca Rubulotta

**Affiliations:** 1Department of Critical Care Medicine, McGill University Health Centre, Montreal, QC H3A 0G4, Canada; 2Department of Critical Care Medicine, University of Catania, 95124 Catania, Italy; francesca.rubulotta@unict.it

**Keywords:** AI, communication, emergencies, research

## Abstract

There is a need to improve communication for patients and relatives who belong to cultural minority communities in intensive care units (ICUs). As a matter of fact, language barriers negatively impact patient safety and family participation in the care of critically ill patients, as well as recruitment to clinical trials. Recent studies indicate that Google Translate and ChatGPT are not accurate enough for advanced medical terminology. Therefore, developing and implementing an ad hoc machine translation tool is essential for bridging language barriers. This tool would enable language minority communities to access advanced healthcare facilities and innovative research in a timely and effective manner, ensuring they receive the comprehensive care and information they need. Method: Key factors that facilitate access to advanced health services, in particular ICUs, for language minority communities are reviewed. Results: The existing digital communication tools in emergency departments and ICUs are reviewed. To the best of our knowledge, no AI English/French translation app has been developed for deployment in ICUs. Patient privacy and data confidentiality are other important issues that should be addressed. Conclusions: Developing an artificial intelligence-driven translation tool for intensive care units (AITIC) which uses language models trained with medical/ICU terminology datasets could offer fast and accurate real-time translation. An AITIC could support communication, and consolidate and expand original research involving language minority communities.

## 1. Background

Language barriers are associated with the risk of misdiagnosis, delayed medical treatment, increased hospital length of stay, poor mental health outcomes among survivors, low patient satisfaction, increased anxiety and depression (as proven by the HADS inventory [1]), and a higher mortality rate [2,3,4]. Recent studies have demonstrated that diversity in intensive care units (ICUs) facilitates communication and results in overall better patient care [5,6,7,8,9]. Although professional interpreter services are effective, there is limited availability in hospitals, especially in time-critical situations in ICUs and out of hours [2,4]. Relatives who assist with translation, when available, may not be able to convey complex medical terminology. Moreover, the stress generated by the critical condition of their loved one might influence their ability to translate [7]. With the rapid development of mobile technology, digital language translation apps can fill this gap. However, recent studies indicate that Google Translate has limited benefit in emergencies and is not accurate enough for advanced medical terminology [2,10]. Moreover, the literature evaluating Google Translate for medical terminology has found that the translation accuracy varies based on the target language [11,12]. One study emphasized that, despite recent reports of improved accuracy and suggestions that Google Translate (GT) could be useful in clinical settings, its accuracy varies significantly across different languages and is not yet dependable for clinical use. Even in languages where GT demonstrates high accuracy, there remains a risk of critical inaccuracies, which could lead to patient harm [13]. According to the British Medical Journal (BMJ) [14], medical defense organizations have advised against doctors using GT during patient consultations due to the risk of communication errors. One study [15] highlights that ChatGPT is not specifically designed or fine-tuned for healthcare applications and should not be used in this context without proper specialization. Moreover, ChatGPT is a proprietary tool, and inputting sensitive patient information to obtain feedback could potentially breach privacy regulations. This review concludes that healthcare researchers should concentrate on advancing NLP research and developing or evaluating specialized language translation tools that are trained on (bio)medical datasets for critical clinical applications.

Translation tools should be prioritized in ICUs over other healthcare settings because of the urgent and critical nature of care required in this environment. Having a translation tool specifically designed for the ICU and trained on ICU datasets is crucial for accurate and fast translation in critical and time-sensitive communication in life-threatening and high-pressure conditions. Such a tool is necessary when dealing with complex and highly specialized ICU terminology in order to reduce the risk of misinterpretation. Misunderstandings due to language barriers can lead to severe consequences, including improper treatment, delayed care, and increased stress for patients and families. Given that patients and families in the ICU are under high emotional and psychological stress, having a precise translation tool tailored for the ICU, could help bridge language barriers and provide them with complex medical information. Moreover, in the ICU, patients often cannot effectively communicate due to their severe medical conditions; however, accurate communication between healthcare providers and patients or their relatives is essential. A specialized translation tool could significantly enhance patient-centered care, improve outcomes and respect the diverse cultural backgrounds of ICU patients. An accurate translation tool would ensure that patients and families can understand informed consent, treatment options and end-of-life care. While translation tools are valuable across all healthcare settings, their impact is most profound in the high-stakes environment of the ICU.

## 2. Rationale

Effective communication between patients, their relatives and medical professionals is essential in the context of complex medical care [2]. This is paramount for treating critically ill patients in the ICU who require sedation and invasive mechanical ventilation [3,4]. As an example, anglophones living in Quebec and francophone patients outside of Quebec face challenges in accessing care in their languages and are at higher risk of poor health [2,16,17,18]. In critical and stressful situations, patients and their relatives who belong to cultural minority communities could lose their ability to convey the extent of their desired treatment, particularly when the care givers do not speak the patient’s primary language. Similarly, healthcare providers and researchers face significant barriers in explaining innovative care or in promoting advanced high technology research. Complex, critical and stressful situations are common in the ICU setting.

## 3. Methods

A thorough search was conducted across relevant databases, including PubMed, Google Scholar, and Scopus, utilizing keywords such as “language minority communities”, “advanced healthcare access”, “ICU”, “digital communication tools”, “AI translation”, “natural language processing in healthcare”, “machine translation in healthcare”, “Good Translate in healthcare” and “ChatGPT in healthcare”. Studies published in the last ten years were prioritized to ensure the inclusion of the most current advancements in technology and methodologies. The authors included articles that discussed the impact of language barriers on healthcare access, the role of digital tools in enhancing communication, and any existing AI-driven solutions relevant to healthcare translation. Studies focused exclusively on non-emergency healthcare settings or lacking empirical data were excluded from this review. Key themes and findings from the selected articles were extracted and organized to identify critical factors influencing access to ICU services for language minority communities. The data were synthesized to highlight trends concerning the efficacy of digital communication tools in overcoming language barriers and to evaluate the current state of AI applications in healthcare translation. Additionally, we assessed prevailing concerns related to patient privacy and data confidentiality in the context of deploying AI translation tools in healthcare, ensuring ethical considerations were part of our analysis.

## 4. Results

To the best of our knowledge, there is currently no English/French AI-driven translation application specifically designed and deployed within ICU settings. This gap significantly impedes timely and effective communication between healthcare providers and patients who speak languages other than the dominant languages of the local healthcare system. Various digital communication tools have been implemented in healthcare environments to facilitate communication between healthcare providers and patients. However, most of these tools lack tailored features for addressing the specific needs of language minority communities. This review identified common barriers such as limited access to qualified interpreters and inadequate multilingual resources within existing systems.

This review also highlighted critical concerns regarding patient privacy and data confidentiality associated with the integration of AI translation tools in healthcare environments. Ensuring that patient information remains secure while utilizing such technologies is paramount to their successful implementation.

Despite existing challenges, the positive trajectory of AI technologies presents a vital opportunity. Developing an artificial intelligence-driven translation tool for intensive care units (AITIC) that utilizes language models trained with a medical/ICU terminology dataset could offer swift and accurate real-time translations.

## 5. Discussion

Machine learning is a powerful and versatile digital tool that enhances healthcare communication by improving patient care and education, allowing for faster decision-making and optimization of resource use, reducing healthcare costs. As examples, natural language processing (NLP), a field of machine learning, and deep neural networks (DNNs), a specific method or architecture within machine learning, have been researched and implemented across different aspects of healthcare. NLP is a branch of AI that enables computers to interact with human (neural) languages. An AI-based chatbot is a virtual assistant or conversational agent which can communicate with patients via text or voice [19,20,21,22]. The chatbot can automatically create and retrieve medical histories and manage scheduling appointments and patient registration [23]. In this study, the application of AI models as translation tools to assist the provider in communicating with patients/family members who speak a different language or have limited health education will be investigated.

Machine translation (MT) focuses on the automated process of converting text from one language to another. MT can be used to either deliver more precise translations or to reduce the time and costs required compared to human translation [24,25].

Panayiotou, A. et al. [11] evaluated 15 iPad-compatible translation apps for the top 10 languages spoken in Australia to assess their suitability for healthcare settings. The results show the low suitability of Google Translate, Microsoft Translator, MediBabble translator, Universal Doctor Speaker, Canopy Speak, SayHi, never Papago, and TripLingo. Among the apps assessed, only two, CALD Assist and Talk To Me, were deemed highly suitable for use in the healthcare setting. Translation technology cannot fully replace a professional interpreter; however, it is neither practical nor financially feasible to provide professional interpreters for every interaction. This highlights the urgency of further study and the need to develop tools that enable safe and effective communication [10]. Rao, P. et al. [26] compared ChatGPT and Google Translate (GT) in their effectiveness at accurately translating instructional and educational medical documents into multiple languages (English into Spanish, Vietnamese, and Russian). While Google Translate remains the most widely used translation tool, ChatGPT has the potential to achieve comparable results. The study indicates that ChatGPT may reliably translate medical information into Spanish but was less effective when translating into other languages. ChatGPT should not be used as a substitute for a professional translator and must be approached with caution due to the high error rate observed in the study. This is particularly crucial for less commonly used languages, where limited data availability may result in lower quality translations [26]. Moreover, GT proved to be of limited value in making medical record content more understandable for patients [27].

It is essential for the scientific community to comprehend ChatGPT’s capabilities and limitations. This involves identifying the specific tasks and domains where ChatGPT is most effective, as well as recognizing potential challenges and constraints [28]. Lundmark, A., & Boglind, F. [29] examine ChatGPT’s performance as a tool for translating a medical terminology dictionary. The study involves translating a Danish medical events dictionary into Swedish using various methods. The findings highlight ChatGPT’s in-context learning capabilities, demonstrating that it can surpass Google Translate when provided with appropriate reference examples [29].

Sarella, P. N. K., & Mangam, V. T. [30] reviewed the benefits and obstacles of integrating AI into healthcare. It becomes evident that AI-NLP is more than a technological tool; it is a transformative force with the potential to revolutionize healthcare communication. By empowering patients with accessible, personalized information, reducing administrative burdens, and supporting diagnosis and treatment planning, AI-NLP is poised to redefine the patient experience and elevate care quality. However, this transformation comes with ethical complexities, including concerns about patient privacy, data security, and algorithmic bias. Addressing these challenges requires a commitment to responsible innovation. Collaboration among healthcare professionals, technology developers, policymakers, and society at large are essential to fully harness AI’s potential while adhering to the highest ethical standards. With thoughtful oversight, transparency, and a focus on equitable, patient-centered care, AI-driven healthcare communication can achieve meaningful and lasting improvements for all [30]. Névéol, A. et al. [31] provides a comprehensive overview of clinical NLP for languages other than English including Indo-European languages such as French, Swedish, and Dutch, as well as Sino-Tibetan (e.g., Chinese), Semitic (e.g., Hebrew), and Altaic (e.g., Japanese, Korean) languages, summarizing recent studies and highlighting opportunities in this growing field. Recent studies indicate that (1) the field is maturing, and (2) researchers now have access to datasets enabling the development of robust methods for tasks such as electronic health record (EHR) de-identification, clinical entity recognition, normalization, and contextualization. However, there is a need for shared tasks and datasets to allow comparative analysis across languages and approaches. Challenges in systematically identifying relevant literature also underscore the importance of adopting more structured publication guidelines that detail language and task specifics. Analyzing language-specific and task-specific nuances could foster methodological innovations in adaptive approaches for clinical NLP [31].

Noack, E.M. et al. [10] designed a digital communication tool to run on a smartphone to help paramedics overcome language barriers when delivering services in the ambulance or in the emergency department. However, no AI methods such as machine learning (ML) or natural language processing (NLP) algorithms were used by these authors [9]. Soto, X. et al. [32] developed a Machine Translation (MT) system for converting clinical text from Basque to Spanish, which is deployed in the Basque public health service.

To the best of our knowledge, no English/French Al translation app has been developed for deployment in ICUs. Patient privacy and data confidentiality are other important issues that an AITIC app project could address. In summary, the proposal to develop an AI machine learning app has the rationale of enabling real-time AI French/English translation for intensive care settings in Quebec. According to Statistic Canada [33], anglophones account for 13.4% of Quebec’s population overall; however, their geographic distribution is highly uneven. Anglophones in the Montréal Census Metropolitan Area (CMA) make up 80.5% (or 801,000 individuals) of Quebec’s total anglophone population. In the Montréal CMA, 22% of the population has English as their first official language spoken (FOLS). A brief analysis of the 2006 Census data shows that anglophones in Quebec are proportionally more likely than francophones to be employed in sectors such as professional, scientific, and technical services, administrative and management services, or wholesale trade. There is a growing need for AI translation tools to facilitate seamless and accurate translation between French and English, ensuring effective communication across linguistic boundaries.

In the future, this can be extended to support additional languages beyond French and English to meet the need of diverse user bases. An AITIC needs to be trained with specialized datasets that include comprehensive medical terminology and context-specific dialogues used in ICU settings. Unlike general translation tools, the AITIC must accurately interpret complex terms, diagnoses, treatment protocols and symptoms, as well as culturally specific language, while maintaining patient safety and confidentiality. Misinterpretations can lead to misunderstandings of symptoms, inaccurate diagnoses, or improper treatment plans, posing serious risks to patient safety. The AITIC must capture the nuances of conversations between providers and patients to ensure effective communication and safe care delivery. Therefore, developing an AITIC, requires collaboration between AI developers and medical professionals to validate datasets and to assess whether the AITIC interprets complex terminology, clinical instructions and nuanced patient descriptions, and ensure the technology is effective and safe for real-world communication in critical medical settings.

Our findings underscore the critical importance of integrating advanced technological solutions, such as AI-driven language translation tools, to bridge communication gaps for language minority communities in accessing advanced health services. By overcoming language barriers, an AITIC could facilitate timely access to critical information and healthcare resources, thus enhancing the quality of care and outcomes for these communities.

Furthermore, the development of an AITIC could significantly contribute to original research that focuses on the experiences and needs of language minority populations in healthcare settings. By ensuring effective communication, an AITIC could support more inclusive health services and empower healthcare providers to deliver culturally competent care tailored to the diverse needs of their patients.

Converting academic research into certified medical devices presents significant challenges, particularly due to the complex regulatory environment surrounding AI in healthcare. Strict requirements for privacy, security, and ethical practices, such as those imposed by The General Data Protection Regulation (GDPR) and the Health Insurance Portability and Accountability Act (HIPAA), demand strong governance and risk management from healthcare organizations. Addressing these challenges requires collaboration among regulatory bodies, industry stakeholders, and healthcare providers. Key obstacles include data quality and availability, lack of standardization, adaptability to clinical settings, trust, acceptance, ethical concerns, and regulatory compliance. Overcoming these barriers is essential to fully realize AI’s potential in enhancing patient outcomes and healthcare delivery [34,35,36].

Addressing ethical considerations around patient privacy and data confidentiality is essential to the successful adoption of AI translation tools in ICUs. Stakeholders must prioritize robust data protection measures and engage in ongoing dialogue with language minority communities to establish trust and transparency in the use of these technologies.

In conclusion, as healthcare continues to evolve, embracing AI-driven solutions tailored to the linguistic and cultural needs of language minority communities will be vital in promoting equitable access to advanced healthcare services. Future research should focus on the practical implementation of an AITIC in ICUs and its efficacy in improving outcomes for patients with language barriers.

## Data Availability

Not applicable.

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
