# Peer review of "Artificial Intelligence-Driven Translation Tools in Intensive Care Units for Enhancing Communication and Research"

_ijerph, 2025, doi:10.3390/ijerph22010095_

Round 1

Reviewer 1 Report

Comments and Suggestions for Authors

Find attached my review report

Author Response

Brief summary

The paper presents a project for developing an MT tool to be deployed in Intensive Care Units in Quebec, Montreal. The main strengths would be the motivation, as visible in the “Background” section when describing the typical healthcare communication, as well as the low resource scenario specified in the “Rationale” section. Furthermore, the methodology is adequate, highlighting the importance of “patient privacy and data confidentiality” in the Abstract, and with a proper ethical approach shown in the “Discussion” section.

Things that could be improved, as specified in the suggestions below, include minor issues related with language clarity and missing citations. Apart from that, the contribution of this paper, even if significant, may be not enough, considering that it seems to describe a recently started project. Nevertheless, the survey work is quite complete, and covers a relevant application of state-of-the-art NLP tools, filling a gap between research and deployment in a critical scenario.

Thank you for your valuable feedback and suggestions. Your thoughtful feedback and constructive comments have been immensely helpful in improving the quality and clarity of the work. Your suggestions have been instrumental in refining the study, and We are confident that the revisions will contribute to a more robust and impactful final publication.

Suggestions

1- Lines 2-3 (clarity): I would suggest to use the term “Machine Translation” (and accordingly, “MTIC”) instead of “Artificial Intelligence-Driven Translation (AITIC)”. In case this suggestion is considered, please apply the corresponding changes to the whole manuscript.

Thank you for your insightful comment. Since the translation tool we describe is primarily NLP-based rather than strictly ML-based, we have chosen to use the term “AI” as it serves as a broader, more inclusive term encompassing various approaches within artificial intelligence, including natural language processing. We believe this terminology accurately reflects the general nature of the technologies involved. We appreciate your understanding and consideration.

2- Lines 2-3 (clarity): besides, I would suggest to include “A project for developing an” in the beginning of the title, so it better describes the content of the paper.

If the journal allows for a longer title, we would be happy to consider revising it accordingly to better reflect the content and scope of the manuscript.

3- General (question): along the paper, sometimes diverse healthcare settings are mentioned, and other times specifically “Intensive Care Units” are referred to. Regarding this, how do you justify the proposal of developing specific MT tools for ICUs, and not for other healthcare scenarios?

Thanks for pointing this out, we have added the following paragraph to the manuscript. This change can be found in background sections. Lines 61-78

4- Line 13 (typo): “Intensive Care units” → “Intensive Care Units”

Agree. We have, accordingly, modified the phrase to emphasize this point. Line 13

5- Line 16 (typo): “Google translate” → “Google Translate” (similarly in Line 115)

Agree. We have edited them. Lines 16, now 148

6- Line 16 (clarity): “not accurate enough for translating advanced medical terminology” (similarly in Line 45)

We have added a paragraph in background to emphasize this point. Lines 47-60

7- Lines 17-18 (clarity and specification): “Artificial Intelligence-driven language translation tool”→ “ad hoc machine translation tool”

Agree. It is corrected. Lines 17-18

8- Lines 23-24 (clarity and correction): “To the best of our knowledge, no machine translation app has been developed specifically for deployment in ICUs.” (similarly in Lines 170-171, correctly specified in Lines 79-80). Justification: the work cited below describes the development of an MTsystem that is currently deployed in the Basque public health service, including its ICUs.

Soto, X., Perez-de-Viñaspre, O., Oronoz, M., & Labaka, G. (2022). Development of a machine

translation system for promoting the use of a low resource language in the clinical domain: The

case of basque. In Natural Language Processing In Healthcare (pp. 139-158). CRC Press.

Thank you for pointing out this. We have added the reference [32] and an explanation in discussion line 203. We have modified the sentence to “no English/French machine translation app has been developed specifically for deployment in ICUs”

9- Line 26 (correction): “AITC” → “AITIC” (or “MTIC”, in case the first suggestion is considered). Anyways, please use a consistent term along the manuscript; or more distinguishable acronyms, in case they refer to two different concepts.

Agree. We applied corrections directly through the text.

10- Line 27 (typo): “dataset” → “ datasets”

Agree. It is corrected. Line 27

11- Lines 35-36 (citation needed): please provide reference (and URL, if applies) for “HADS

inventory”.

We have added the citation. Lines 35-36

12- Line 47 (typo): “.” missing in the end of the sentence.

Agree. It is corrected. Line 47

13- Methods (question): how were the search terms mentioned in Lines 62-64 selected? Which criteria or previous work were considered to include these terms and not others?

We selected these search terms to ensure our review articles comprehensively cover key topics, with a particular focus on the use of AI and translation tools in healthcare and we also added additional terms to fully encompass our research criteria in Methods section (lines 94-97).

14- Lines 99-101 (typos and correction): “Natural language processing (NLP) and deep neural

network (DNN) are two field of machine learning” → “Natural Language Processing (NLP) and

Deep Learning (DL) are two fields of machine learning” Justification for the latter: deep neural

network refers to a method or architecture, more than a field of machine learning.

Agree. It is corrected. Now Lines 134-135

15- Line 102 (clarity): “understand” → “interact with” Justification: currently there is an open debate whether computers “understand human languages” or not. Furthermore, the use of the term “understand” related to computers contributes to anthropomorphize machines, which can have negative outcomes, especially in healthcare settings.

Agree. It is corrected. Now Line 137

16- Lines 104-106 (clarity): I would suggest to remove the text “be developed with combination of Natural Language Processing, Computer Vision, and Machine Learning and are able to”.

We have removed the text.  Now lines 137-138

17- Line 109 (typo): “speaks” → “speak”

Agree. It is corrected. Now line 142

18- Line 119 (typo): “However” → “however”

Agree. It is corrected. Now line 152

19- Line 120 (typo): “urgent” → “ urgency”

Agree. It is corrected. Now line 153

20- Line 121 (typo): “need” → “a need” (also, missing “.” in the end of the sentence after “[10]” at Line 122.)

Agree. It is corrected.  Now line 154

21- Line 173 (missing context): even if widely known, sociolinguistic studies mentioning the

percentage of speakers of “French/English” (and other languages) in Quebec would help to provide the necessary context to any reader.

Agree. We have added a reference from Statistic Canda regarding Official-Language Minorities in Canada - Anglophones in Quebec. The explanations can be found in the discussion. lines 209-219.

22- Line 174 (clarity): “Quebec, this can be extended” → “Quebec. In the future, this can be

extended”

Agree. It is corrected. Now line 220

23- Line 175 (typo): “required” → “requires”

Agree. It is corrected. Now line 221

24- Line 179 (typo): “confidentially” → “confidentiality”

Agree. It is corrected. Now line 225

25- Line 185 (typo): “clinal” → “clinical”

Agree. It is corrected. Now line 231

26- General (clarity): please be consistent with the citations and adhere to the journal specifications. In this regard, sometimes full names are provided (for example, in Line 113), while other times just acronyms are given (for example, in Line 153).

Agree. We made the modifications in the manuscript.

27- General (clarity): I would suggest to use the term “relatives” instead of “family members”, to include friends and others.

Agree. We applied corrections directly through the text.

Once again, thank you for your invaluable contribution to the review process. Your dedication and commitment to advancing scholarly research are truly appreciated.

Best regards,

Sahar Bahrami

Francesca Rubulotta

Reviewer 2 Report

Comments and Suggestions for Authors

Interesting topic.  Need to strengthen the background information. 

Author Response

Thank you for your valuable feedback and suggestions. Your suggestions have been instrumental in refining the study, and We are confident that the revisions will contribute to a more robust and impactful final publication.

  • Abbreviation stands for?

AITIC stands for Artificial Intelligence-Driven Translation Tool in Intensive Care Unit, we have added this to the Abstract. Line 26

  • In the abstract, you include ChatGPT has not being sufficient for translating medical information. However, in your introduction, you do not talk about ChatGPT or provide any evidence for why this statement is in the abstract.

Thanks for pointing this out. We have added an explanation of the reasons why ChatGPT is not sufficient for translating medical terminology to the introduction. Lines 47-60

  • Google

Agree. We have corrected: google translate -> Google Translate, line 44

  • Evidence for ChatGPT. Some of this needs to be discussed in the introduction/background.

We have added an explanation of the reasons why ChatGPT is not sufficient for translating medical terminology to the introduction. Lines 47-60

  • 2 sentences or combined--punctuation

We have edited this: “Sarella, P. N. K., & Mangam, V. T. [30] reviewed the benefits and obstacles of integrating AI into healthcare. It becomes evident that AI-NLP is more than a technological tool”: They are two sentences. it -> It , lines 174-175

Once again, thank you for your invaluable contribution to the review process. Your dedication and commitment to advancing scholarly research are truly appreciated.

Best regards,

Sahar Bahrami

Francesca Rubulotta
